# Impact of Persistent Inflammation, Immunosuppression, and Catabolism Syndrome during Intensive Care Admission on Each Post-Intensive Care Syndrome Component in a PICS Clinic

**DOI:** 10.3390/jcm12165427

**Published:** 2023-08-21

**Authors:** Shinya Suganuma, Masafumi Idei, Hidehiko Nakano, Yasuaki Koyama, Hideki Hashimoto, Nobuyuki Yokoyama, Shunsuke Takaki, Kensuke Nakamura

**Affiliations:** 1Department of Critical Care Medicine, Yokohama City University Hospital, 3-9 Fukuura, Kanazawa-ku, Yokohama 236-0004, Japan; suganuma.shi.en@yokohama-cu.ac.jp (S.S.); idei.mas.bn@yokohama-cu.ac.jp (M.I.); voth@yokohama-cu.ac.jp (N.Y.); shunty5323@gmail.com (S.T.); 2Department of Emergency and Critical Care Medicine, Hitachi General Hospital, 2-1-1, Jonan-cho, Hitachi 317-0077, Japan; be.rann1988jp@gmail.com (H.N.); yasukoya@md.tsukuba.ac.jp (Y.K.); hidehashimoto-tky@umin.ac.jp (H.H.)

**Keywords:** critical care, persistent inflammation, immunosuppression, catabolism syndrome, PIICS, post-intensive care syndrome, PICS, PICS clinic

## Abstract

Background: Persistent inflammation, immunosuppression, and catabolism syndrome (PIICS) is known as a prolonged immunodeficiency that occurs after severe infection. Few studies have demonstrated a direct relationship between PIICS and physical dysfunction in post-intensive care syndrome (PICS). We herein investigated how each component of PICS was affected by the diagnosis of PIICS during hospitalization and examined the relationship between PIICS and PICS using PICS assessments performed at the Hitachi General Hospital PICS Clinic. Methods: The 273 patients who visited the PICS clinic at one month after discharge from the ICU at Hitachi General Hospital were included in the study. We used the diagnostic criteria for PIICS described in previous studies. At least two of the following blood test values on day 14 of hospitalization had to be met for a diagnosis of PIICS: C-reactive protein (CRP) > 2.0 mg/dL, albumin (Alb) < 3.0 g/dL, and lymphocytes (Lym) < 800/μL. Blood test values closest to day 14 out of 11–17 days of hospitalization were used. The primary outcome was a Barthel Index (BI) < 90, while secondary outcomes were the results of various PICS assessments, including mental and cognitive impairments, performed at the PICS clinic. We supplemented missing data with multiple imputations by chained equations. We performed a nominal logistic regression analysis with age, sex, BMI, SOFA, and the presence of PIICS as variables for BI < 90. Results: Forty-three out of two hundred seventy-three PICS outpatients met the diagnostic criteria for PIICS during hospitalization. In comparisons with non-PIICS patients, significantly higher severity scores for APACHE II and SOFA and a longer hospital stay were observed in PIICS patients, suggesting a higher clinical severity. The primary outcome, BI, was lower in the PIICS group (97.5 (58.5, 100) vs. 100 (95, 100), *p* = 0.008), as were the secondary outcomes (FSS-ICU: 35 (31, 35) vs. 35 (35, 35), MRC score: 55 (50.25, 58) vs. 58 (53, 60), grip strength: 16.45 (9.2, 25.47) vs. 20.4 (15.3, 27.7)). No significant differences were noted in mental or cognitive function assessments, such as HADS, IES-R, and SMQ. A multivariable analysis supplemented with missing data revealed that PIICS (odds ratio: 1.23 (1.08–1.40 *p* = 0.001) and age (odds ratio: 1.007 (1.004–1.01), *p* < 0.001) correlated with BI < 90, independent of clinical severity such as sequential organ failure assessment (SOFA). Similar results were obtained in the sensitivity analysis excluding missing data. Conclusions: The present study revealed a strong relationship between PIICS and post-discharge PICS physical dysfunction in patients requiring intensive care.

## 1. Introduction

Survival rates for life-threatening conditions have improved, and post-intensive care syndrome (PICS), a complication after treatment has become a problem [1]. PICS comprises physical, cognitive, and mental impairments that occur during admission to or after discharge from the intensive care unit (ICU) and worsen the long-term prognosis of ICU patients [1]. The incidence of PICS is estimated to be more than 50% [2]. PICS requires a long-term follow-up on an outpatient basis after discharge [3].

Persistent inflammation, immunosuppression, and catabolism syndrome (PIICS) is a prolonged immunodeficiency that occurs after severe infection [4,5]. Many patients with PIICS exhibit a strong inflammatory response, cachexia due to catabolism and malnutrition, poor wound healing, and increased susceptibility to infection [6]. Long-term mortality is elevated in patients with PIICS due to increased susceptibility to infection and the worsening of activities of daily living (ADL) [4,5].

Although PICS and PIICS both have a significant impact on the prognosis of post-intensive care patients, their relationship currently remains unclear. PIICS, an immunological sequela, is expected to be a physiological risk factor for PICS because catabolism is included in the concept. A previous study [7] suggested a relationship between PIICS and physical dysfunction in PICS. On the other hand, the relationships between PIICS and the physical, mental, and cognitive functions assessed in a PICS clinic setting after hospital discharge remain unknown.

PICS clinics are not common, particularly in Asian countries [8,9,10]. Japan’s first PICS clinic opened at Hitachi General Hospital in 2019, treating critically ill patients, including all ICU patients [11]. Various tests performed at the PICS clinic have provided novel insights into the relationship between physical and mental dysfunctions [11]. We conducted a retrospective study to elucidate the relationships between PIICS during hospitalization and each PICS component in the PICS clinic.

## 2. Materials and Methods

### 2.1. Ethics Approval and Informed Consent

This study is a single-center retrospective study of the relationship between PICS assessment and PIICS in the PICS Clinic at Hitachi General Hospital (651 beds, 8 ICU beds, Ibaraki, Japan). The present study was approved by the Ethics Committee on February 23, 2018 (Approval No. 2017-95).

### 2.2. Patient Selection

The PICS clinic opened in August 2019 and was opened every Thursday. All ICU patients and patients admitted to the Emergency and Critical Care Center were instructed to visit the PICS Clinic one month after discharge. The following is an overview of the ICU at the Emergency Center of Hitachi General Hospital. There is a medical and surgical ICU (8 beds) with one nurse for every two patients to handle severely conditioned patients, including postoperative patients, and an emergency ward (10 beds) with one nurse for every four patients. Patients discharged from the medical and surgical ICUs and who stayed in the emergency ward for more than five days were referred to the PICS clinic approximately one month after discharge. Brochures were used to explain the PICS clinic to patients and patients’ families. Only the first visit to the PICS clinic was included; return visits were excluded. PICS clinic visits were not performed if the patient did not wish to be seen. Patients who visited the PICS clinic for the first time between August 2019 and April 2022 were included in the study.

### 2.3. Outcomes and Measurements

ICU physicians, nurses, and physical therapists assessed each component of PICS (physical, cognitive, and mental status) to diagnose PICS. Physical, cognitive, and mental status at the PICS clinic was assessed as follows. At the PICS clinic, physicians evaluated physical dysfunction by assessing gait disorder, muscle volume loss, and respiratory dysfunction. Patients were judged to have gait disturbance if they had difficulty walking 50 m on a level ground compared to before admission to the ICU. They also assessed mental dysfunction by the presence of depression, anxiety, and sleep disorders. Cognitive dysfunction was assessed by the presence of executive dysfunction and memory impairment. Physical performance was examined by physiotherapists using the following parameters: the Barthel Index (BI) [12], the functional status score for ICU (FSS-ICU) [13], the Medical Research Council (MRC) score [14], and grip strength (kg) in the left and right hands and its mean. The Hospital Anxiety and Depression Scale (HADS) [15] and Impact of Event Scale-Revised (IES-R) [16] were evaluated by nurses using a questionnaire as measures of mental status and post-traumatic stress disorder, respectively. The Short-Memory Questionnaire (SMQ) [17] was used by nurses as a measure of cognitive function; however, it was not possible to evaluate some patients with deficits in cognitive functions. Figure 1 shows a schematic diagram of the PICS components, evaluation items, and evaluators at the Hitachi General Hospital PICS Clinic.

The diagnosis of PIICS was based on blood test values during hospitalization. According to a previous study [18], at least two of the following blood test values on day 14 of hospitalization had to be met for a diagnosis of PIICS: CRP > 2.0 mg/dL, Alb < 3.0 g/dL, and Lym < 800/μL. Based on the findings of a previous study [18], blood test values closest to day 14 out of 11–17 days of hospitalization were used. Patients without blood tests after day 11 of hospitalization and those discharged after less than 11 days of hospitalization were treated as non-PIICS.

Other patient backgrounds, such as age, the SOFA score, Acute Physiology and Chronic Health Evaluation II (APACHE II) score, BMI, and length of hospital stay, were also retrospectively investigated.

The primary outcome was physical impairment defined as BI < 90. BI < 90 was defined as a state of decreased ADL and need for assistance, in accordance with previous studies [19,20]. Secondary outcomes were each diagnostic item of physical, mental, and cognitive impairments.

### 2.4. Statistical Analysis

We compared the PIICS and non-PIICS groups. Continuous variables were described as means ± standard deviation. If the null hypothesis was not rejected by the Shapiro–Wilk test, comparisons were made using Welch’s *t*-test. If the null hypothesis was rejected by the Shapiro–Wilk test, continuous variables were expressed as medians (interquartile range) and compared using the Mann–Whitney U test. Non-parametric paired values were expressed as medians (interquartile ranges) and compared using the Wilcoxon signed-rank sum test. The categorical variables were calculated as the percentage of patients in each category. We performed a univariate analysis to identify parameters associated with BI < 90. Age, gender, BMI, and severity of illness, commonly known as risk factors for PICS [21,22], were added as explanatory variables in the analysis.

A nominal logistic regression analysis was conducted with age, sex, BMI, SOFA, and the presence of PIICS as variables for BI < 90. We supplemented missing data with multiple imputations by chained equations (MICE). We performed a complete case analysis excluding missing data as a sensitivity analysis. Results with a *p*-value < 0.05 were indicated with * and considered to be significant. All statistical analyses were conducted using stats models of Python 3.

## 3. Results

Figure 2 shows the study outline. A total of 3946 patients were admitted to the Emergency and Critical Care Center; 543 died during admission and 3403 were discharged alive. Among these patients, 786 patients stayed in the medical and surgical ICU for ≥1 day or in the emergency ward for ≥5 days and PICS clinic reservations were made. In total, 273 patients visited the PICS clinic one month after hospital discharge. Forty-three out of two hundred seventy-three patients were diagnosed with PIICS and analyzed in the present study.

Table 1 shows the baseline clinical data of patients. Table 2 shows the baseline clinical data of patients and PICS clinic examination results for the PIICS and non-PIICS groups. In comparisons with non-PIICS patients, significantly higher severity scores for APACHE II (median: 17 [IQR: 13.5, 20.5] vs. 13 [9, 17.25), *p* < 0.001]) and SOFA (7 [5, 8.5] vs. 5 [3, 7], *p* < 0.001) and a longer hospital stay (18 [13, 29.75] vs. 6 [3, 10], *p* < 0.0001) were observed in PIICS patients, suggesting a higher clinical severity. The primary outcome, BI, was lower in the PIICS group (median: 97.5 [IQR: 58.5, 100] vs. 100 [95, 100], *p* = 0.0076), as were the secondary outcomes (FSS-ICU: 35 [31, 35] vs. 35 [35, 35], MRC score: 55 [50.25, 58] vs. 58 [53, 60], grip strength: 16.45 [9.2, 25.47 vs. 20.4 [15.3, 27.7]). Among the 223 patients in the non-PIICS group, 150 (67.26%) had a perfect BI score, 175 (78.83%) out of 222 had a perfect FSS-ICU score, and 100 (45.25%) out of 221 had a perfect MRC score. None of the following mental and cognitive test results were significantly different between the PIICS and non-PIICS groups: total HADS, HADS (depression), HADS (anxiety), total IES-R, IES-R (intrusion), IES-R (avoidance), IES-R (hyperarousal), and total SMQ.

Table 3 shows the results of the single regression analysis and nominal logistic regression analysis with BI < 90 and each explanatory variable. When the objective variable was BI < 90, significant differences were observed in age (odds ratio: 1.06 [1.03–1.09], *p* < 0.001) and PIICS (odds ratio: 2.79 [1.36–5.73], *p* = 0.005), but not in BMI, SOFA, APACHE II, or blood test values. Based on the results of the single regression analysis, age, sex, BMI, SOFA, and PIICS were selected as explanatory variables in the logistic regression analysis. The multivariable analysis supplemented with missing data showed that PIICS (odds ratio: 1.23 [1.08–1.40], *p* = 0.0012) and age (odds ratio: 1.007 [1.004–1.01], *p* < 0.001) were associated with BI < 90, independent of severity. Sex, SOFA, and BMI were not associated with BI < 90. Similar results were obtained in the complete case analysis of 169 subjects excluding missing data, which was conducted as a sensitivity analysis.

## 4. Discussion

The present study showed that PIICS during hospitalization was associated with physical dysfunction in PICS clinic visits after discharge. Although a relationship between PIICS and physical dysfunction was previously suggested [7], the present study was the first to assess each PICS component in the PICS clinic and use optimized diagnostic criteria for PIICS.

The following mechanism is proposed to be responsible for the development of PIICS. Stress, such as trauma and infection, induces the production of inflammatory cytokines and TNFα, which promote muscle catabolism. Hematopoietic stem cells are also activated and emergency myelopoiesis occurs. During this process, myeloid-derived suppressor cells (MDSC) develop in damaged tissue [5]. MDSC exert immunosuppressive effects through various mechanisms, including the induction of T-cell apoptosis. MDSC may also cause persistent inflammation through their ability to produce inflammatory mediators, nitric oxide (NO), and reactive oxygen and may contribute to nerve and muscle tissue damage [5]. These mechanisms have been suggested to delay the recovery of physical dysfunction in patients with PIICS.

The present study showed no relationships between PIICS and the mental and cognitive function components of PICS. Physical dysfunction quickly appears in the acute phase of intensive care and subsequently improves. In contrast, mental dysfunction often gradually worsens after discharge from the hospital, and a time lag has been reported [23,24]. Therefore, the relationships between PIICS and the mental and cognitive function components of PICS may be weak in the near-acute period, namely, one month after discharge, which was the target of the present study. Another reason why this relationship was not observed in this patient population may be because younger patients are more prone to mental disorders [22]. The mean age of patients in the present study was 70 years, and many patients were elderly. Among critically ill patients, older patients are at a higher risk of developing cognitive dysfunction [25,26], which may have diminished the impact of PIICS. However, the long-term relevance of this issue is currently unknown.

There is no evidence to support the amelioration of the symptoms of each PICS component by PICS clinic interventions [27]. However, an evaluation of each component of PICS in a PICS clinic and continued follow-ups as a high-risk group for physical dysfunction are recommended [3]. Therefore, patients diagnosed with PIICS during hospitalization require more careful follow-ups in a PICS clinic. Nutritional and exercise therapies may be useful as therapeutic interventions for PIICS [28,29]. Patients with PIICS show hyper catabolism and degradation of nutrients including protein, carbohydrates, and lipids. These are speculated to involve inflammatory cytokines such as IL-6, IL-1, and TNFα, as well as intestinal dysfunction. Catabolism is intricately intertwined with immunosuppression and persistent inflammation to form the pathogenesis of PIICS. Nutritional therapy may be useful in ameliorating intestinal dysfunction and hyper catabolism and may benefit PIICS patients [28]. Although the effects of exercise therapy on PIICS patients are unknown, early rehabilitation in the ICU is associated with improved physical function, and the combination of exercise and nutritional therapy outside the ICU may be useful in improving protein synthesis and muscle strength. Exercise therapy may also be beneficial for patients with PIICS [29]. In a PICS clinic, nutrition and exercise therapies in consideration of PIICS may improve PIICS and, ultimately, PICS.

It is important to note that some PIICS patients have severe PICS and, thus, are unable to visit a PICS clinic. A previous study [30] investigated PIICS in patients with disseminated intravascular coagulation (DIC). Clinical severity scores such as SOFA and APACHE Ⅱ for DIC patients and ADL were worse in these patients than in our patient population, and PICS was also more severe. Therefore, considering follow-up approaches for PIICS patients who cannot visit the PICS clinic due to severe PICS is a future issue.

There are several limitations that need to be addressed. This was a single-center retrospective study and the results obtained may have been affected by center-dependent treatment policies and patient populations in the ICU. Furthermore, an assessment bias in the PICS clinic cannot be ruled out. However, a wide range of patient groups were included in the present study: 48.72% internal medicine, 50.55% surgery, and 85.35% admitted from the emergency room. We also employed widely accepted objective scoring to eliminate these biases. Another limitation is that since only ICU patients who visited the PICS outpatient clinic were included in the analysis, the present results may not be applicable to all ICU patients. Patients with severe physical dysfunction may have had difficulty coming to the PICS clinic, and many patients with mild or moderate disease may have been included in the analysis. Moreover, patients with low cognitive function may have been included in the present study because cognitive function was not assessed prior to admission. To overcome this problem with external validity, we plan to conduct a prospective multicenter study. In addition, although we identified patients with PIICS based on blood test values on days 11–17 of hospitalization, some patients who were discharged less than 11 days after admission may also have met the diagnostic criteria for PIICS. Although PIICS is characterized by immunosuppression, blood test results alone may not be sufficient to detect it. Therefore, we adopted what we considered to be the best diagnostic criteria based on previous studies [18]. It is possible that a superior diagnostic method will be developed as our understanding of PIICS increases.

## 5. Conclusions

The present results revealed a strong relationship between PIICS diagnosed during the hospitalization of patients requiring intensive care and physical dysfunction in the PICS clinic after discharge. On the other hand, a relationship was not observed between PIICS and the mental or cognitive function components of PICS. A more detailed understanding of PIICS and the development of effective treatments for PIICS are expected to prevent and attenuate PICS.

## Figures and Tables

**Figure 1 jcm-12-05427-f001:**
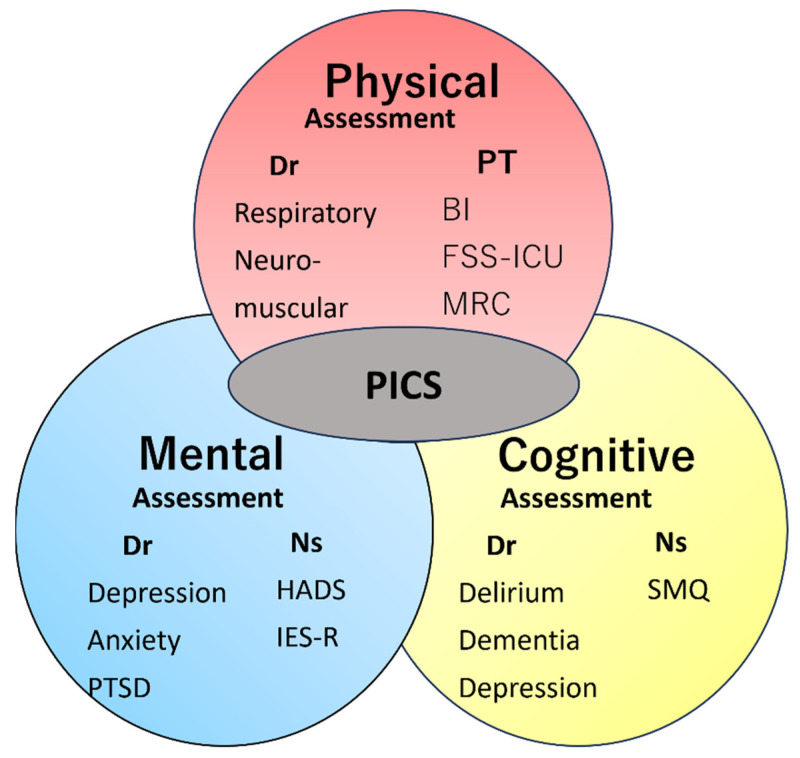
Schematic diagram of physical, mental, and cognitive assessment items and assessors at the Hitachi General Hospital PICS Clinic.

**Figure 2 jcm-12-05427-f002:**
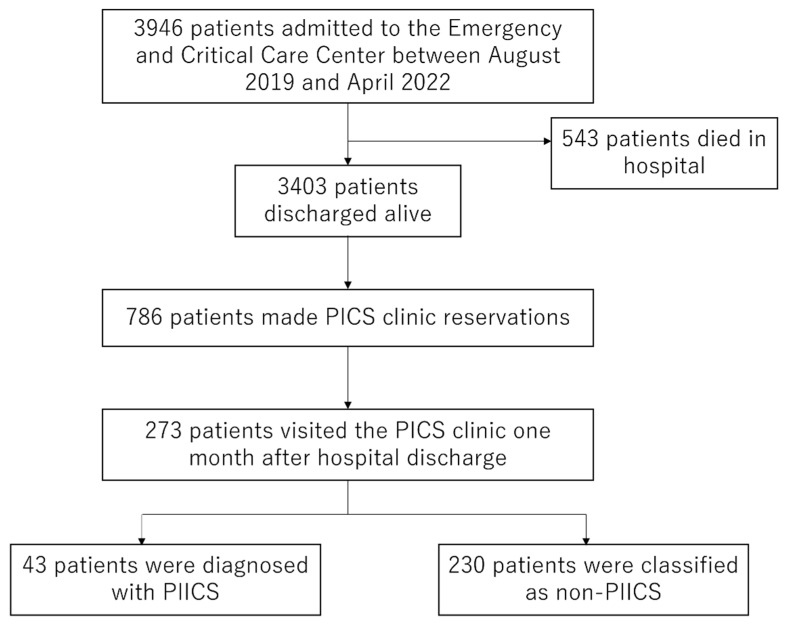
Study outline. ICU, intensive care unit; PICS, post-intensive care syndrome; PIICS, persistent, inflammation, immunosuppression, and catabolism syndrome.

**Table 1 jcm-12-05427-t001:** Baseline characteristics on admission to the ICU.

	N	273
Admission data	Age	70.82 (±14.14)
	Sex (male)	168 (61.54%)
	BMI	23.63 (±4.4)
	SOFA	5 (3, 8)
	APACHE Ⅱ	14 (9, 18)
	Length of hospital stay	7 (3, 13)
	Internal medicine	133 (48.72%)
	General surgery	138 (50.55%)
	Emergency room	233 (85.35%)
	Postoperative patient	11 (4.03%)
	11–17-day C-reactive protein	1.58 (0.20, 7.92)
	11–17-day albumin	2.6 (2.15, 3.1)
	11–17-day total lymphocyte count	1347 (971, 1781)

SOFA, sequential organ failure assessment score; APACHE II, acute physiology, and chronic health evaluation. Age, BMI: mean ± standard deviations. Sex, internal medicine, general surgery, emergency room, postoperative patient: percentages of patients. Others: medians (interquartile ranges).

**Table 2 jcm-12-05427-t002:** PIICS vs. non-PIICS. Baseline characteristics on admission to the ICU and PICS outcomes at hospital discharge and in the PICS clinic.

		PIICS	Non-PIICS	*p*-Value
	n	43	230	Value
Admission data	Age	68.93 (±25.36)	71.17 (±28.08)	0.31
	Sex (male)	28 (65.12%)	140 (60.87%)	0.60
	BMI	23.66 (±8.31)	23.63 (±8.69)	0.97
	SOFA	7 (5, 8.5)	5 (3, 7)	<0.001 *
	APACHE Ⅱ	17 (13.5, 20.5)	13 (9, 17.25)	<0.001 *
	Length of hospital stay	18 (13, 29.75)	6 (3, 10)	<0.001 *
	11–17-day C-reactive protein	6.26 (3.19, 9.66)	0.88 (0.45, 1.33)	<0.001 *
	11–17-day albumin	2.3 (1.9, 2.6)	3.05 (2.6, 3.3)	<0.001 *
	11–17-day total lymphocyte count	1092 (591, 1555)	1482 (1165, 1809)	0.0026 *
At the PICS clinic				
Physical status	Barthel Index	97.5 (58.5, 100)	100 (95, 100)	0.0076 *
	FSS-ICU	35 (31, 35)	35 (35, 35)	0.008 *
	MRC score	55 (50.25, 58)	58 (53, 60)	<0.001 *
	Grip strength (kg)	16.45 (9.2, 25.47)	20.4 (15.3, 27.7)	0.01 *
Mental status	Total HADS	8 (5, 14)	10 (5.5, 15)	0.53
	HADS (depression)	5 (3, 8)	6 (3, 10)	0.57
	HADS (anxiety)	3 (1, 7)	3 (1, 6)	0.68
	Total IES-R	5 (2, 11)	5 (2, 13)	0.93
	IES-R (intrusion)	2 (1, 4)	2 (1, 5)	0.99
	IES-R (avoidance)	1 (0, 6)	1 (0, 4)	0.34
	IES-R (hyperarousal)	1 (0, 4)	1.5 (0, 4)	0.34
Cognitive status	Total SMQ	38 (26, 43)	37.5 (28.5, 42)	0.99

Results with a *p*-value ≦ 0.05 are indicated with * and are considered to be significant. FSS-ICU, functional status score for the ICU; MRC, Medical Research Council score; HADS, Hospital Anxiety, and Depression Scale; IES-R, Impact of Event Scale-Revised; SMQ, Short-Memory Questionnaire.

**Table 3 jcm-12-05427-t003:** Univariate/multivariable logistic regression analyses of BI <90. Results with a *p*-value < 0.05 were indicated with * and considered to be significant.

BI < 90	Univariate Logistic Regression Analysis		Multivariable Logistic Regression Analysis (MICE)		Multivariable Logistic Regression Analysis (Complete Case Analysis)	
	Odds Ratio (95% CI)	*p*-Value	Odds Ratio (95% CI)	*p*-Value	Odds Ratio (95% CI)	*p*-Value
Age	1.06 (1.03−1.09)	<0.001 *	1.007 (1.004−1.010)	<0.001 *	1.07 (1.03−1.11)	0.001 *
Sex	0.68 (0.37−1.24)	0.21	1.048 (0.857−1.031)	0.19	1.01 (0.45−2.25)	0.99
SOFA	1.00 (0.90−1.11)	0.99	1.008 (0.984−1.014)	0.92	1.03 (0.90−1.67)	0.70
BMI	0.95 (0.87−1.04)	0.24	1.006 (0.982−1.009)	0.60	0.97 (0.87−1.08)	0.53
APACHE Ⅱ	0.99 (0.96−1.03)	0.65				
CRP day 1	1.02 (0.99−1.05)	0.31				
CRP days 11–17	1.03 (0.94−1.13)	0.54				
Alb days 11–17	0.51 (0.24−1.08)	0.076				
Lym days 11–17	1.01 (0.98−1.05)	0.50				
PIICS	2.79 (1.36−5.74)	0.005 *	1.23 (1.08−1.40)	0.0012 *	2.85 (1.05−7.71)	0.04 *

BI, Barthel Index; SOFA, sequential organ failure assessment score; APACHE II, acute physiology and chronic health evaluation; CRP, C-reactive protein; Alb, albumin; LYM, lymphocyte count; PIICS, persistent, inflammation, immunosuppression, and catabolism syndrome; 95% CI, 95% confidence interval; MICE, multiple imputation by chained equations.

## Data Availability

The datasets generated and analyzed during the present study are available from the corresponding author upon reasonable request.

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
