# Peer review of "Impact of Persistent Inflammation, Immunosuppression, and Catabolism Syndrome during Intensive Care Admission on Each Post-Intensive Care Syndrome Component in a PICS Clinic"

_jcm, 2023, doi:10.3390/jcm12165427_

Round 1
Reviewer 1 Report
The authors studied the relationship between Persistent Inflammation, Immunsuppression, and Catabolism Syndrome (PIICS) and Post Intensive Care Syndrome (PICS). The study utilized a PICS clinic that is not common. They showed a strong relationship between PIICS and post-discharge PICS but no relationship between PIICS and the mental/cognitive function components of PICS.
I found this to be a novel study that will be of interest to the readership of the journal. There are a few edits that need to be addressed:
1. Abbreviations need to be spelled out when first introduced in the manuscript. For example, CRP, Alb, Lym (line 34).
2. The legend of Table 2 can easily be condensed since it basically describes what is in the table. With the scores, references, and abbreviations it is unreadable.
English quality is acceptable.
Author Response
Dear. Reviewer
Journal of Clinical Medicine
We would like to thank you for your careful review and comments related to manuscript ID jcm-2540174 entitled “Impact of Persistent Inflammation, Immunosuppression and, Catabolism Syndrome during Intensive Care Unit stays on each Post Intensive Care Syndrome assessment in the PICS clinic.”. We are sending the revised manuscript with point-by-point responses to the comments provided by the Reviewers. We thank all the Reviewers for their meaningful and valuable comments.
With best regards,
Shinya Suganuma, M.D.
Responses to the Reviewers
REVIEWER REPORTS
Reviewer Comments:
Reviewer 1
Major Comments:
Abbreviations need to be spelled out when first introduced in the manuscript. For example, CRP, Alb, Lym (line 34).
Our answer: We have revised the text as you indicated.
The legend of Table 2 can easily be condensed since it basically describes what is in the table. With the scores, references, and abbreviations it is unreadable.
Our answer: Thank you for your suggestion. The results in Table 2 have been briefly described.
A revised manuscript with change track is attached.
Thank you very much in advance.

Reviewer 2 Report
The study provides valuable insights into the relationship between PIICS and PICS, particularly the physical dysfunction component. The findings may have significant implications for the care and rehabilitation of ICU survivors, and the group should be commended for driving research within the area of PIICS.
I have but a few notes:
Title
1. The comma before Catabolism is unnecessary and just confusing.
2. Could "Intensive Care Unit stays" be simplified to e.g. "intensive care admission" or similar?
3. Is it really the impact of PIICS on PICS assessment, and not PICS components you are examining?
4. "the PICS clinic". Which one? I think the "the" confuses the reader.
Abstract
5. Line 26. "Widely". Is it really widely known? I think PIICS is rather absent from most discussions?
6. Line 37. Since MICE is not introduced as an abbreviation here, I don't think Multiple should be capitalised.
7. Line 46. I think the authors should reevaluate whether they are using multivariate or multivariable analysis.
8. Line 47. In general, I think results are using way too many decimals.
9. Line 48. Severity of what?
Introduction
10. Line 55. I find it a bit strange to argue that there's a recent increase by citing a 12-year-old article, published at the same time as PICS as a concept was introduced?
11. Line 66. ADL should perhaps be explained?
12. Line 78. Most "herein" are unnecessary.
13. Line 79. See #3.
Results
14. Line 151-154. I don't get the formatting here. Are these supposed to be single lines?
15. Line 177. Parenthesis within parenthesis should be bracketed? ([])
16. Line 182 (Table 1). Please rewrite the variables with correct capitalisation (I don't mean BMI, SOFA etc. but age, sex etc.)
17. Line 182 (Table 1). Values in parentheses should be explained. Are they SD, percentages, etc?
18. Line 187 (Table 2). See #8.
19. Line 188. It should be p less than or equal, not just less than.
20. Line 208. NO should be explained.
Discussion
21. Line 227-228. This is more repetitive than reasoning and could perhaps be rewritten and expanded a bit since it is such a central part of the Discussion.
22. Line 232. "The severity of DIC patients and ADL..." Severity of DIC patients? Severity of what? Severity of ADL? This sentence is a bit confusing.
23. Line 247. Overcome external validities? Don't you rather overcome problems with external validity?
Running the text through Grammarly, something I personally always do, being a non-native English writer, I (well, Grammarly) found about 50 recommendations for change. Not all were relevant, but I think the text could be somewhat improved with some more editing.
Author Response
Dear. Reviewer
Journal of Clinical Medicine
We would like to thank you for your careful review and comments related to manuscript ID jcm-2540174 entitled “Impact of Persistent Inflammation, Immunosuppression and, Catabolism Syndrome during Intensive Care Unit stays on each Post Intensive Care Syndrome assessment in the PICS clinic.”. We are sending the revised manuscript with point-by-point responses to the comments provided by the Reviewers. We thank all the Reviewers for their meaningful and valuable comments.
With best regards,
Shinya Suganuma, M.D.
REVIEWER REPORTS
Reviewer Comments:
Reviewer 2
Major Comments:
The comma before Catabolism is unnecessary and just confusing.
Could "Intensive Care Unit stays" be simplified to e.g. "intensive care admission" or similar?
"the PICS clinic". Which one? I think the "the" confuses the reader.
Our answer: We have revised the above as you indicated above.
Is it really the impact of PIICS on PICS assessment, and not PICS components you are examining?
Our answer: That's right - a study of PICS components in PICS clinics and their relationship to PIICS.
Line 26. "Widely". Is it really widely known? I think PIICS is rather absent from most discussions?
Our answer: I think you are correct. I have changed the word "widely known".
Line 37. Since MICE is not introduced as an abbreviation here, I don't think Multiple should be capitalised.
Our answer: We have revised the text as you indicated.
Line 46. I think the authors should reevaluate whether they are using multivariate or multivariable analysis.
Our answer: Thank you for pointing this out. I have revised it to "multivariable" because there is only one result variable.
Line 47. In general, I think results are using way too many decimals.
Our answer: We have revised the points you pointed out.
Line 48. Severity of what?
Our answer: Severity" refers to the clinical severity of the disease. The relevant section has been revised.
Line 55. I find it a bit strange to argue that there's a recent increase by citing a 12-year-old article, published at the same time as PICS as a concept was introduced?
Our answer: Thank you for pointing this out." I meant "in recent years" rather than "recently". I have changed the wording of the sentence.
Line 66. ADL should perhaps be explained?
Line 78. Most "herein" are unnecessary.
Our answer: Thank you for pointing this out. We have revised it.
Line 151-154. I don't get the formatting here. Are these supposed to be single lines?
Our answer: It appears to be a transcription error. It has been deleted.
Line 177. Parenthesis within parenthesis should be bracketed?
Line 182 (Table 1). Please rewrite the variables with correct capitalisation (I don't mean BMI, SOFA etc. but age, sex etc.)
Line 182 (Table 1). Values in parentheses should be explained. Are they SD, percentages, etc?
Line 188. It should be p less than or equal, not just less than.
Line 208. NO should be explained.
Our answer: Thank you for pointing this out. We have revised it.
Line 227-228. This is more repetitive than reasoning and could perhaps be rewritten and expanded a bit since it is such a central part of the Discussion.
Our answer: Thank you very much. I have added the text as it is a central part of the discussion as you mentioned.
Line 232. "The severity of DIC patients and ADL..." Severity of DIC patients? Severity of what? Severity of ADL? This sentence is a bit confusing.
Our answer: Sorry, I meant the clinical severity of DIC patients. I have rewritten the text for clarity.
Line 247. Overcome external validities? Don't you rather overcome problems with external validity?
Our answer: You are correct. We have revised it.
A revised manuscript with change track is attached.
Thank you very much in advance.
King regards,
Shinya Suganuma.
